# *Tcf4* Is Involved in Subset Specification of Mesodiencephalic Dopaminergic Neurons

**DOI:** 10.3390/biomedicines9030317

**Published:** 2021-03-20

**Authors:** Simone Mesman, Iris Wever, Marten P. Smidt

**Affiliations:** Swammerdam Institute for Life Sciences, FNWI, University of Amsterdam, 1098 XH Amsterdam, The Netherlands; iris.wever@nijgeerten.nl

**Keywords:** midbrain development, dopamine neurons, *Tcf4*, subset specification, bHLH factors

## Abstract

During development, mesodiencephalic dopaminergic (mdDA) neurons form into different molecular subsets. Knowledge of which factors contribute to the specification of these subsets is currently insufficient. In this study, we examined the role of *Tcf4*, a member of the E-box protein family, in mdDA neuronal development and subset specification. We show that *Tcf4* is expressed throughout development, but is no longer detected in adult midbrain. Deletion of *Tcf4* results in an initial increase in TH-expressing neurons at E11.5, but this normalizes at later embryonic stages. However, the caudal subset marker *Nxph3* and rostral subset marker *Ahd2* are affected at E14.5, indicating that *Tcf4* is involved in correct differentiation of mdDA neuronal subsets. At P0, expression of these markers partially recovers, whereas expression of *Th* transcript and TH protein appears to be affected in lateral parts of the mdDA neuronal population. The initial increase in TH-expressing cells and delay in subset specification could be due to the increase in expression of the bHLH factor *Ascl1*, known for its role in mdDA neuronal differentiation, upon loss of *Tcf4*. Taken together, our data identified a minor role for *Tcf4* in mdDA neuronal development and subset specification.

## 1. Introduction

Mesodiencephalic dopaminergic (mdDA) neurons are thought to arise from neural progenitors in the ventricular zone (VZ) of the floor plate (FP) and basal plate (BP) of the midbrain from E11.5 onward [1,2,3]. When neural progenitors acquire an mdDA neuronal cell fate, they migrate toward their final location and gain a subset-specific identity [4,5,6,7,8]. Of these subsets, the neuroanatomically distinct substantia nigra (SNc) and ventral tegmental area (VTA) are best known, mainly due to the selective degeneration of the SNc during Parkinson’s disease (PD) [9]. The specification of the different subsets in the mdDA system is dependent on a unique set of transcription factors and signaling cascades during development [4,5,6,7,8,10], although knowledge of the factors that are important during subset specification is currently insufficient.

One protein family important in neuronal differentiation is the basic helix–loop–helix (bHLH) family of transcription factors, an enormous protein family that can be subdivided into different subfamilies [11]. The function of bHLH proteins differs throughout embryonic development and is dependent on their spatial–temporal expression pattern and binding partner(s) [12,13]. Several members of the bHLH protein family, e.g., *Ascl1* (*Mash1*), *Ngn2*, *Tcf12*, and *Nato3*, have been shown to be involved in early stages of mdDA neuronal development during fate decision of neural progenitors and later subset specification [14,15,16,17].

The E-box protein family is a small subfamily of the bHLH protein family, and contains 3 family members; *Tcf3* (*E2A*), *Tcf4* (*E2-2*), and *Tcf12* (*Heb*) [18], which are expressed throughout the developing embryo and embryonic brain. These proteins are known to specifically bind to a so-called E-box (CANNTG) in the promoter of different genes, and can act as an activating or repressive factor via homo- and heterodimerization with other bHLH factors [12,13]. Most studies of the function of E-box proteins focus on their role in immune-system development. They have been found to be critical for the differentiation from CD4^+^CD8^+^ double-positive T cells to either CD4^+^ or CD8^+^ single-positive T cells [19]. However, each E-box protein plays a unique role in immune-system development, and loss of *Tcf4* results in a milder phenotype than loss of *Tcf12* or *Tcf3* [19,20].

During brain development, E-box proteins are thought to regulate common targets for neurogenesis and unique target genes for neuronal subtype development, dependent on their interaction partner and expression area [13]. Fischer et al. [21] have shown that E-box proteins may function together with *Ascl1* to fine-tune the differentiation progression in the subventricular zone of the adult forebrain during postnatal neurogenesis. Furthermore, loss of *Tcf12* and *Tcf3* leads to an overall reduction in brain size, although the morphology of brain structures is not altered [22], and loss of *Tcf12* during midbrain development has been shown to result in a delay in mdDA neuronal differentiation and ultimately affected subset development [16]. *Tcf4* has been implicated in several neurodevelopmental diseases, like autism and schizophrenia, and loss of one copy of *Tcf4* leads to major brain malformations resulting in Pitt–Hopkins Syndrome in humans [23,24]. During murine brain development, *Tcf4* has been shown to have a pivotal role in development of the cortex, hippocampus, pontine nucleus, and major axonal tracts like the corpus callosum and anterior commissure [25,26,27].

In this study, we focus specifically on the role of *Tcf4* in the development of the mdDA neuronal system. In the rat brain, TCF4 has been shown to interact with CDP2 (the rat homologue for *Cux* genes in mice) [28]. This interaction is thought to relieve the repression of CDP2 on the *Th* gene, thereby initiating *Th* transcription [28]. Contrary to what was detected in rat, we did not find a decrease in TH expression in the mice *Tcf4* mutant midbrain during development. We detected a transient upward trend in the amount of TH-expressing cells at E11.5, which was no longer apparent at E12.5 and normalized at E14.5. However, expression of the subset specific markers *Ahd2* (rostral subset) and *Nxph3* (caudal subset) was strongly affected at this developmental stage, which partly recovered during terminal differentiation. It is possible that the early increase in the amount of TH^+^ cells in the midbrain and affected expression of subset markers is due to upregulation of the bHLH factor *Ascl1* upon loss of *Tcf4*. Taken together, we have established the E-box protein *Tcf4* as a novel factor involved in correct development of mdDA neurons.

## 2. Experimental Section

### 2.1. Ethics Statement

All animal studies were performed in accordance with local animal-welfare regulations, as this project was approved by the animal experimental committee (Dier ethische commissie, Universiteit van Amsterdam; DEC-UvA), and international guidelines.

### 2.2. Animals

The transgenic mouse line B6;129-Tcf4^tm1Zhu^/J originated from the Jackson Laboratory. B6;129-Tcf4^tm1Zhu^/J mouse line was back-crossed with the C57BL/6 line and WT and mutant littermate embryos to study the effects of the loss of *Tcf4* on mdDA neuronal development, were generated by crossing heterozygous *Tcf4* mice. To study the WT expression pattern of *Tcf4*, WT embryos were generated by crossing C57BL/6 mice. Pregnant mice (embryonic day 0.5 (E0.5) is defined as the morning of plug formation) were sacrificed by cervical dislocation. Embryos were collected in 1× PBS and immediately frozen on dry ice, or fixed by immersion of 3–12 h in 4% paraformaldehyde (PFA) at 4 °C. After PFA incubation, samples were cryoprotected O/N at 4 °C in 30% sucrose. Embryos were frozen on dry ice and stored at −80 °C. Cryosections were sliced at 16 μm, mounted on Superfrost Plus slides (Thermo Fisher Scientific, Waltham Massachussetts, USA), air-dried, and stored at −80 °C until further use.

### 2.3. Genotyping

Genotyping of B6;129-Tcf4^tm1Zhu^/J transgenic embryos and mice was performed according to the protocol of the Jackson Laboratory. Briefly, 100 ng of genomic DNA was used together with primer pair: FP 5′-AGCGCGAGAAAGGAACGGAGGA-3′, RP1 5′-GGCAATTCTCGGGAGGGTGCTT-3′, and RP2 5′-CCAGAAAGCGAAGGAGCA-3′, resulting in a product at 229 bp for the WT allele and a product at 400 bp for the KO allele.

### 2.4. In Situ Hybridization and Combined TH-DAB IHC

In situ hybridization with digoxigen (DIG)-labeled probes was performed as described previously [29]. Fresh frozen sections were fixed in 4% PFA for 30 min and acetylated with 0.25% acetic anhydride in 0.1 M triethanolamine for 10 min. Probe hybridization was carried out at 68 °C O/N with a probe concentration of 0.4 ng/μL in a hybridization solution containing 50% deionized formamide, 5× SSC, 5× Denhardt’s solution, 250 μg/mL tRNA Baker’s yeast, and 500 μg/mL sonificated salmon sperm DNA. The following day, slides were washed in 0.2× SSC for 2 h at 68 °C followed by blocking with 10% HIFCS in buffer 1 (100 mM Tris Hcl, pH 7.4 and 150 mM NaCl) for 1 h at RT. DIG-labeled probes were detected by incubating with an alkaline-phosphatase-labeled anti-DIG antibody (Roche, Mannheim, Germany, 1:5000), using NBT-BCIP as a substrate. If in situ hybridization was not followed by TH-DAB immunohistochemistry, slides were washed 2 × 5 min in T_10_E_5_, dehydrated with ethanol, and embedded in Entellan.

DIG in situ hybridization was performed with the following probes: 918 bp *Tcf4* fragment bp 1101–2018 of mouse cDNA, 873 bp *Ascl1* fragment bp 1452–2325 of mouse cDNA, 491 bp *Th* fragment bp 252–143 of mouse cDNA, 1792 bp *Ahd2* fragment bp 5–1801 of mouse cDNA, and 480 bp *Nxph3* fragment bp 1374–1854 of mouse cDNA.

After DIG in situ hybridization, *Tcf4* stained WT sections were used for DAB immunohistochemistry on TH. Slides were incubated in 0.3% H_2_O_2_ in Tris-buffered saline (TBS) for 30 min at RT. Thereafter, blocking was performed with 4% HIFCS in TBS. Slides were incubated O/N with primary antibody Rb-TH (Pel-Freez, Rogers Arkansas, USA 1:1000) in TBS. The following day, slides were incubated for 1 h with goat–anti-rabbit biotinylated secondary antibody (Vector Laboratories, Burlingame, California, USA 1:1000) in TBS, followed by incubation with avidine–biotin–peroxidase reagents (ABC elite kit, Vector Laboratories, Burlingame, CA, USA 1:1000) in TBS. Slides were stained with DAB (3,3′-diamino-benzidine) for a maximum of 10 min. Slides were dehydrated with ethanol and embedded with Entellan.

### 2.5. Immunohistochemistry

Fluorescence immunohistochemistry (IHC) was carried out as described previously [30,31]. Cryosections were blocked with either 4% HIFCS or 5% normal donkey serum (for sheep primary antibodies) in 1× THZT and incubated with a primary antibody (Rb-TH (Pel-Freez, Rogers Arkansas, USA 1:1000), Sh-TH (Millipore, Waltham, MA, USA AB1542, 1:1000), Rb-AHD2 (Abcam, Cambridge, UK, AB24343, 1:200), Rb-PITX3 ([32] 1:1000)), diluted in 1× THZT O/N at RT. The next day, sections were incubated with a secondary Alexafluor antibody (anti-rabbit, anti-sheep) diluted 1:1000 in 1× TBS for 2 h at RT. After immunostaining, nuclei were stained with DAPI (1:3000) and washed extensively in 1× PBS. Slides were embedded in Fluorsave (Calbiochem, Amsterdam, The Netherlands) and analyzed with the use of a fluorescent microscope (DM5500B Leica, Amsterdam, The Netherlands). The antibody against AHD2 required antigen retrieval as follows. Slides were incubated with 0.1 M citrate buffer pH6 for 3 min at 800 W and 9 min a 400 W, then cooled down to RT in a water bath, after which the protocol was followed as described above.

Quantification of TH- and AHD2-expressing cells at embryonic stages was performed as follows. TH- and AHD2-expressing cells were counted in 5–11 (matching) sagittal sections (n = 3 WT; n = 3 KO). Cells were counted as TH^+^ or AHD2^+^ when staining co-localized with a nuclear DAPI staining. Statistical analysis was performed via a one- or two-tailed Student’s *t*-test.

### 2.6. Quantitative PCR (qPCR)

RNA was isolated from dissected E14.5 midbrain of *Tcf4* WT and *Tcf4* mutant embryos (littermates; at least two litters were used per experiment). Relative RNA expression levels were determined by qPCR real-time PCR (Lightcycler 480, Roche, Bazel, Switzerland) using the Quantitect SYBR Green PCR Lightcycler Kit (Qiagen, Hilden, Germany) according to the manufacturer’s protocol. For each reaction, 10 ng (dissected material) total RNA was used as input. Primer sequences are listed in Appendix A.

Quantified qPCR results represent average values of experiments performed on 4 biological samples for each genotype. The qPCR experiments required multiple litters to obtain a sufficient sample size. Measurements that differed by ≥2× standard deviation from the mean were excluded. Statistical analysis was performed via a two-tailed Student’s *t*-test.

## 3. Results

### 3.1. Tcf4 Is Expressed from E11.5 Onward and Partly Overlaps with the TH-Expressing Population

Since bHLH factors are known to be important in mdDA neuronal development and the E-box protein family is expressed throughout the embryonic brain, we investigated the expression pattern of the E-box factor *Tcf4* (*E2-2*) during development in the brain. *Tcf4* was found to be expressed throughout the hindbrain, midbrain, and prosomere 1–3 during development (Figure 1A; anatomic maps in Figure 1A,B were adapted from the Allen Brain Atlas (http://atlas.brain-map.org, accessed on 3 January 2019). At E11.5, *Tcf4* showed a very low level of expression in the midbrain area, and its expression pattern was restricted to the ventricular zone (VZ) of both the floor plate (FP) and basal plate (BP) (Figure 1B). This expression pattern continued at E12.5, and overlap was detected between the *Tcf4* transcript and TH protein in the FP area of the midbrain (Figure 1B). At E14.5, the expression of *Tcf4* increased and became more restricted to the rostral area in both the lateral and medial sections (Figure 1A), although *Tcf4* showed a strong expression in the medial areas of the midbrain (Figure 1B). Although *Tcf4* is expressed in the midbrain during embryonic development, it can no longer be detected in the adult midbrain. Taken together, the expression profile of *Tcf4* suggests that it may have a role during early development of the mdDA neuronal population, but is not likely to have a role in adult mdDA neurons, as its expression was not detected at this stage.

### 3.2. Loss of Tcf4 Leads to an Upward Trend in TH-Expressing Neurons at E11.5 but Not at E14.5, and Sporadic Loss of PITX3 at E14.5

Above we have shown that *Tcf4* is expressed throughout the midbrain during development and is specifically expressed in the VZ at early embryonic stages. MdDA neurons are shown to originate in the VZ of the FP and FP–BP boundary from E10.5 onward, with neuronal birth peaking between E11 and E12 [1,2,33]. To determine whether *Tcf4* has a role in the early cell-fate commitment of neural progenitors toward an mdDA neuronal fate, we performed immunohistochemistry on TH at E11.5 *Tcf4* mutant embryos [34] (Figure 2A, left panel). In the *Tcf4* mutant midbrain, we detected an upward trend in TH-expressing neurons compared to WT embryos (Figure 2A, right panel; WT n = 3, mutant n = 5; *p* = 0.06, one-tailed), suggesting an accelerated neuronal differentiation at E11.5.

To determine whether this accelerated neuronal differentiation early during development is persistent in later stages, we examined the anatomical build-up of the mdDA neuronal population at E12.5 and E14.5. The composition of the mdDA neuronal population already appeared to be normalized at E12.5 (Appendix A), and TH expression in the midbrain of the *Tcf4* mutant at E14.5 was not visibly altered compared to its WT littermates (Figure 2B). Quantification of the TH-expressing neuronal population confirms that the amount of TH^+^ neurons was similar between the *Tcf4* WT (n = 3) and mutant (n = 3) embryos at E14.5. Although the TH-expressing population showed a normal anatomical build-up, in some *Tcf4* mutant embryos, we detected a clear loss of PITX3, a factor specifically expressed in postmitotic dopamine neurons, in the midbrain, although PITX3 expression was still present in the musculature of these embryos (Figure 2C). This effect on PITX3 was only observed in a subset of animals at E14.5 (out of 3 embryos, expression of PITX3 was completely lost in 1 and partially lost in 2 embryos compared to their WT littermates), and was not observed at E13.5 and E15.5 (Figure 2C; n = 3 for all stages). Taken together, these data indicate that loss of *Tcf4* results in a possible accelerated mdDA neuronal differentiation and a not fully penetrating effect on terminal differentiation.

### 3.3. Correct mdDA Neuronal Subset Specification Is Affected in the Tcf4 Mutant

Above we have shown that loss of *Tcf4* appears to have a possible effect on the start of mdDA neuronal differentiation, but does not clearly affect the TH-expressing population at later stages. However, expression of *Tcf4* in specific areas of the midbrain overlapping with mdDA neurons and the possible effect on mdDA neuronal differentiation could point to a function in later differentiation and even result in a change in subset specification.

To determine whether *Tcf4* may have a role in subset development of the mdDA neuronal population, we examined the expression levels of several rostral and caudal subset markers in E14.5 dissected midbrain material by means of qPCR (Figure 3A). Expression of the caudal markers *Tle3* and *Cck* [8] was unaffected in the mdDA neuronal population, whereas the caudal marker *Nxph3* [35] showed a decrease of ~50% in the *Tcf4* mutant midbrain (n = 4; *p* = 0.036, two-tailed). Expression of the rostral markers *Rspo2*, *Pbx3*, *Tle4*, *Ebf1*, and *Ebf2* were unaffected in the *Tcf4* mutant midbrain (n = 4) [8,36]. Expression of *Rspo2* appeared to be downregulated with ~25%, but due to large variation between embryos, this was not significant (n = 4; *p* = 0.35, two-tailed). *Ahd2* expression, on the other hand, showed a downregulation of ~75% in the *Tcf4* mutant (n = 4; *p* = 0.0008, two-tailed). The loss of *Nxph3* in caudal areas and *Ahd2* in rostral areas was confirmed via in situ hybridization at E14.5 (Figure 3B). Similar to *Ahd2* transcript expression, AHD2 protein levels were strongly affected at E14.5 (Figure 3B).

At E15.5, some expression of AHD2 protein was detected, but loss of this protein was still detectable (Figure 3C). Quantification of AHD2-expressing neurons at E15.5 showed a decrease of ~30% in the mutant (WT n = 3, mutant n = 4; *p* = 0.006, one-tailed). Similar to what was detected at E14.5, the amount of TH-expressing neurons was not altered at E15.5 (WT n = 3, mutant n = 4), although the expression of subset-specific markers was clearly altered. These data showed that, although TH-expressing neurons appear to be present in normal numbers at this stage in the *Tcf4* mutant, the expression of specific rostral and caudal subset markers was affected upon loss of *Tcf4*.

### 3.4. Ahd2 and Nxph3 Expression Partially Recovers Later in Development, Whereas Th Expression Becomes Slightly Affected at P0

As shown above, loss of *Tcf4* results in an initially changed subset specification as exemplified by the change in *Ahd2* and *Nxph3* levels. To examine whether the expression of these subset markers remains affected or recovers during mdDA development, we investigated the expression of *Th*, *Ahd2*, and *Nxph3* at P0 (Figure 4). Although the expression of TH was not significantly altered at E14.5, the expression of *Th* at P0 appeared to be affected in the most lateral regions of the mdDA neuronal pool (Figure 4A, black arrowheads). Similar to the expression of *Th* transcript, the expression of TH protein appeared to be affected in the most lateral regions of the mdDA neuronal population (Figure 4A, white arrowheads). However, quantification of the TH-expressing cells in the SNc, VTA, and the total number of TH-expressing cells in the midbrain area shows that the amount of TH-expressing cells was not significantly altered (Figure 4A). Expression of *Ahd2* was strongly decreased at E14.5, but recovered during embryonic development. At P0, the expression of *Ahd2* was almost completely recovered, although some loss could still be detected in the most lateral regions of the mdDA neuronal population (Figure 4B, black arrowheads). *Nxph3* showed a weak expression in the medial–caudal area of the mdDA neuronal system at P0, which was comparable to its expression at E14.5. However, similar to *Ahd2,* its expression appeared to be recovered in the mutant midbrain at P0, although in the most caudal sections expression of *Nxph3* was still slightly affected (Figure 4B, black arrowheads). Pseudo-overlays of adjacent sections of the expression patterns of *Th* and *Ahd2*, and of *Th* and *Nxph3*, showed that the expression of these markers overlapped in both WT and mutant brains, and that the loss of these markers was detected in similar areas (white arrowheads). Together, these data indicated that the expression of subset markers partially recovered during mdDA development. Even more so, expression of TH, which was not affected at E14.5, appeared to be decreased in the most lateral regions of the mdDA neuronal system at P0.

### 3.5. Ascl1 Is Upregulated in the Developing Midbrain of the Tcf4 Mutant

As shown above, loss of *Tcf4* leads to an increase in the amount of TH-expressing neurons in the embryonic midbrain, which has no significant effect on the total TH-expressing population at E14.5, and a loss of expression of the subset markers *Ahd2* and *Nxph3*, which partially recovers during development. It is known from other cell systems that loss of E-box proteins may be compensated by other E-box or bHLH proteins, and that these proteins are involved in the regulation of expression of each other [13,15]. To determine whether the initial increase in TH expression and recovery of *Ahd2* and *Nxph3* expression could be caused by the upregulation of other bHLH factors present in the midbrain during development, we performed qPCR analyzes to examine the expression levels of *Tcf3*, *Tcf12*, *Ngn2*, and *Ascl1* on dissected midbrain material at E14.5 (Figure 5A). Quantitative PCR on *Tcf4* confirmed that *Tcf4* was indeed lost in the *Tcf4* mutant (n = 4; *p* = 0.001, two-tailed). Expression of *Tcf3*, *Tcf12*, and *Ngn2* was not significantly altered in the midbrain of the *Tcf4* mutant, indicating that if these genes compensated for *Tcf4* loss of function, this was not controlled by upregulation of transcription of these factors. However, expression of *Ascl1* was detected to be increased ~1.8-fold in the mutant midbrain at E14.5 (n = 4; *p* = 0.05, two-tailed).

To determine whether *Tcf4* expression overlaps with the expression of *Ascl1*, we performed in situ hybridization on E12.5 and E14.5 WT material. *Tcf4* expression strongly overlapped in the VZ of the midbrain at E12.5 with *Ascl1* expression (Figure 5B). At E14.5, the expression of *Tcf4* increased throughout the midbrain, whereas the expression of *Ascl1* diminished at this stage. Pseudo-overlays of adjacent sections of *Tcf4* and *Ascl1* expression showed the overlap in expression at these developmental stages. Interestingly, at E14.5 the overlap with the expression of *Ascl1* seemed to decrease in the areas of the midbrain where *Tcf4* was strongly expressed, suggesting a possible reciprocal expression at later stages. To examine whether *Tcf4* normally inhibits the expression of *Ascl1* or whether the loss of *Tcf4* is compensated by an upregulation of *Ascl1,* we performed in situ hybridization for *Ascl1* on E12.5 and E14.5 WT and mutant littermates (Figure 5C). *Ascl1* showed an expanded expression pattern at both E12.5 and E14.5 in the VZ of the mutant midbrain (Figure 5C, black arrowheads). Together, these data indicated that the loss of *Tcf4* resulted in an increase in expression of *Ascl1,* and that *Tcf4* either regulated the expression of *Ascl1*, or that the loss of *Tcf4* may have been partly compensated by an upregulation of this bHLH factor in the embryonic midbrain at the stages analyzed.

## 4. Discussion

Proneural bHLH factors like *Ngn2*, *Ascl1*, *Tcf12*, and *Nato3* are known to be important in neuronal specification in the midbrain [14,15,16,17]. However, besides these bHLH factors, other members of the bHLH protein family of transcription factors are also expressed in the developing midbrain, like *Tcf4*, a member of the bHLH subfamily of E-box proteins. In this study, we focused on the role of *Tcf4* in mdDA neuronal development. *Tcf4* is expressed throughout the midbrain during development, but is no longer detected in the adult midbrain. During development, its expression is present in the VZ and overlaps with the rostral part of the TH-expressing population.

Loss of *Tcf4* leads to an initial upward trend in the amount of TH-expressing cells early during development, which is recovered at E14.5, indicating that, different from what was detected in the rat [28], *Tcf4* is not of major importance in the regulation of TH-expression in mdDA neurons in mice midbrain. However, as E-box binding sites are present in the promoter of the human, mouse, and rat *Th* gene, it may be of interest to determine whether TCF4 is able to bind to the promoter in the mouse brain and fine-tune *Th* expression in developing mdDA neurons. Interestingly, in a subset of embryos at E14.5, a change in expression of PITX3 protein was detected, which was not seen in earlier or later stages of development. The fact that this phenotype was only detected in some embryos and was not persistent throughout development suggests that the loss function of *Tcf4* is not fully penetrating and may be influenced by genetic background and/or compensation by other bHLH factors.

Although the anatomical build-up of the mdDA neuronal domain seems relatively normal, we found that both *Nxph3*, a marker for the caudal mdDA subset, and *Ahd2*, a maker for the rostral mdDA subset, were changed in the *Tcf4* mutant midbrain, indicating that the mdDA neuronal population is generated, but the identity of the different subsets is affected in the absence of *Tcf4*. These data suggest that *Tcf4* has a specific role in subset specification. As the expression of both *Nxph3* and *Ahd2* is mostly recovered at P0, loss of *Tcf4* leads to a delayed subset specification. This regulation of subset identity is likely established in the VZ of the midbrain, or when mdDA neurons migrate through *Tcf4* expressing areas toward their final location, as the expression of *Tcf4* does not completely overlap with the expression areas of *Nxph3* and *Ahd2*.

From earlier studies, it is known that bHLH factors can compensate for each other and regulate each other’s expression in the embryonic midbrain [14,15]. Here, we detected that *Ascl1* showed a significant upregulation during development as a consequence of *Tcf4* ablation. *Ascl1* is known to be important in the onset of mdDA neuronal differentiation [14,15]. The increase in expression of *Ascl1* may possibly lead to a premature differentiation of neural progenitors into mdDA neurons, linking *Tcf4* to the correct timing of mdDA neuronal differentiation. However, the delay in expression of the subset markers *Ahd2* and *Nxph3* suggests that the premature differentiation of TH-expressing neurons is only partial. Interestingly, although the expression of *Ahd2* and *Nxph3* is mostly recovered at P0, the expression of *Th* transcript and TH protein is visibly altered in lateral areas of the mdDA neuronal population, suggesting that the initial changed subset specification results in a permanent change in the mdDA neuronal population. As *Tcf4* has been shown to affect axonal outgrowth and development of the major axonal tracts [27], it is possible that *Tcf4* affects the projection of TH-expressing neurons to the striatum, eventually resulting in the degeneration of a subset of TH+ neurons, as they cannot establish the correct connections.

In the present study, we have established *Tcf4* as a novel factor involved in the differentiation and specification of the mdDA neuronal population. Loss of *Tcf4* may lead to premature neuronal differentiation, which has no apparent effect on the anatomical build-up and amount of TH-expressing cells at E14.5, although subset specification is initially affected. Interestingly, *Tcf4* affects expression of *Nxph3*, a marker that was previously shown to be lowered in PD patients and an important factor in the survival of DA neurons derived from induced pluripotent stem cells, making *Tcf4* an possible target in search of stem-cell-based therapies for PD [37].

## Figures and Tables

**Figure 1 biomedicines-09-00317-f001:**
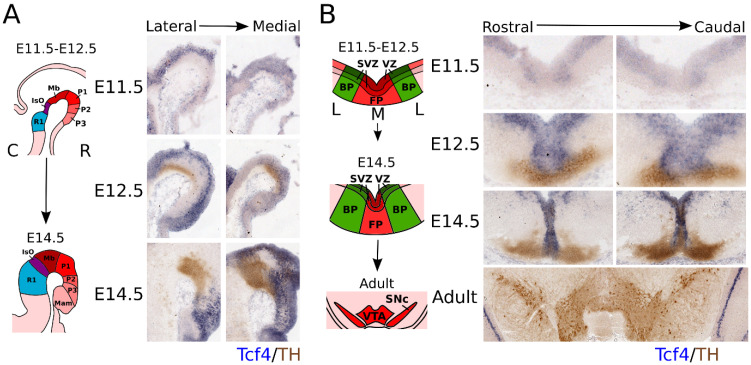
*Tcf4* is expressed in the VZ of the developing midbrain and overlaps with part of the rostral TH-expressing population. (**A**) Sagittal view of *Tcf4* expression throughout development. *Tcf4* (blue) is expressed from E11.5 onward in the developing embryonic midbrain area. At E11.5, expression of *Tcf 4*. is specific for the VZ from R1–P3. At E12.5, its expression overlaps with the medial part of the TH-expressing neuronal population (brown), which is more apparent at E14.5 and even strongly overlaps with the rostral TH^+^ population. Anatomic maps were adapted from the Allen Brain Atlas (http://atlas.brain-map.org accessed on 3 January 2019). C: caudal, R: rostral. (**B**) Coronal view of *Tcf4* expression throughout development and in the adult. *Tcf4* (blue) shows expression in the VZ of the BP of the midbrain at E11.5. Expression in the FP is not clearly detected until E12.5. Overlap with the TH-expressing population (brown) can mainly be detected in medial parts of the midbrain at E12.5 and E14.5. In the adult midbrain, *Tcf4* is no longer detected. Anatomic maps were adapted from the Allen Brain Atlas (http://atlas.brain-map.org accessed on 3 January 2019). L: lateral, M: medial.

**Figure 2 biomedicines-09-00317-f002:**
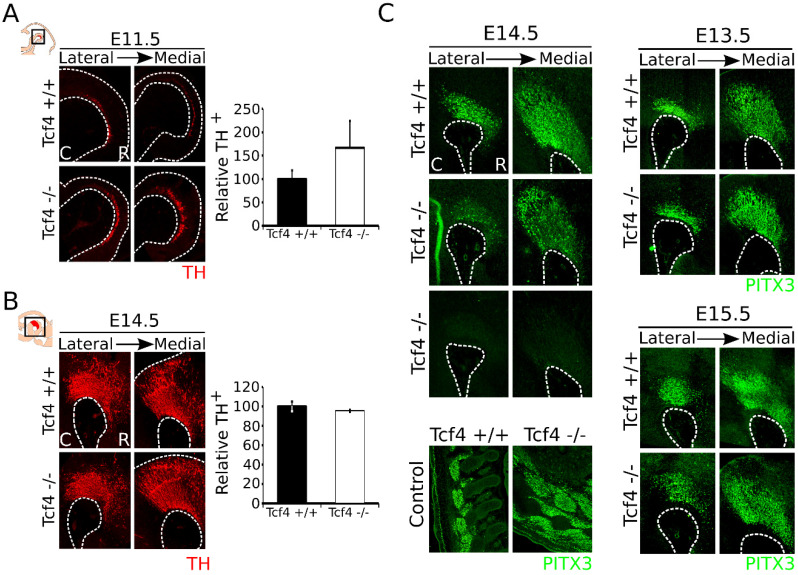
The *Tcf4* mutant shows an upward trend in the amount of TH^+^ cells at E11.5, which normalizes at E14.5. (**A**) Loss of *Tcf4* leads to an upward trend in TH-expressing cells (red) in the mutant embryo at E11.5. Quantification of the TH^+^ neurons shows an upward trend in *Tcf4* mutant embryos (white bar) compared to WT embryos (black bar), although this is not significant (WT n = 3; mutant n = 5; *p* = 0.06 one-tailed). WT was set at 100%. C: caudal, R: rostral. (**B**) Loss of *Tcf4* has no apparent effect on the morphology and the presence of TH-expressing neurons (red) in the midbrain at E14.5. Quantification of TH^+^ neurons in the WT (black bar) and *Tcf4* mutant (white bar) midbrain does not show a difference in the amount of the TH-expressing neurons (n = 3; two-tailed). WT was set at 100%. C: caudal, R: rostral. (**C**) Expression of PITX3 (green) is affected in some *Tcf4* mutant embryos at different levels in the embryonic midbrain at E14.5. This effect on PITX3 was not seen in the musculature of the tested embryos. Furthermore, at E13.5 and E15.5 no clear difference in PITX3 (green) expression level or area was detected. C: caudal, R: rostral.

**Figure 3 biomedicines-09-00317-f003:**
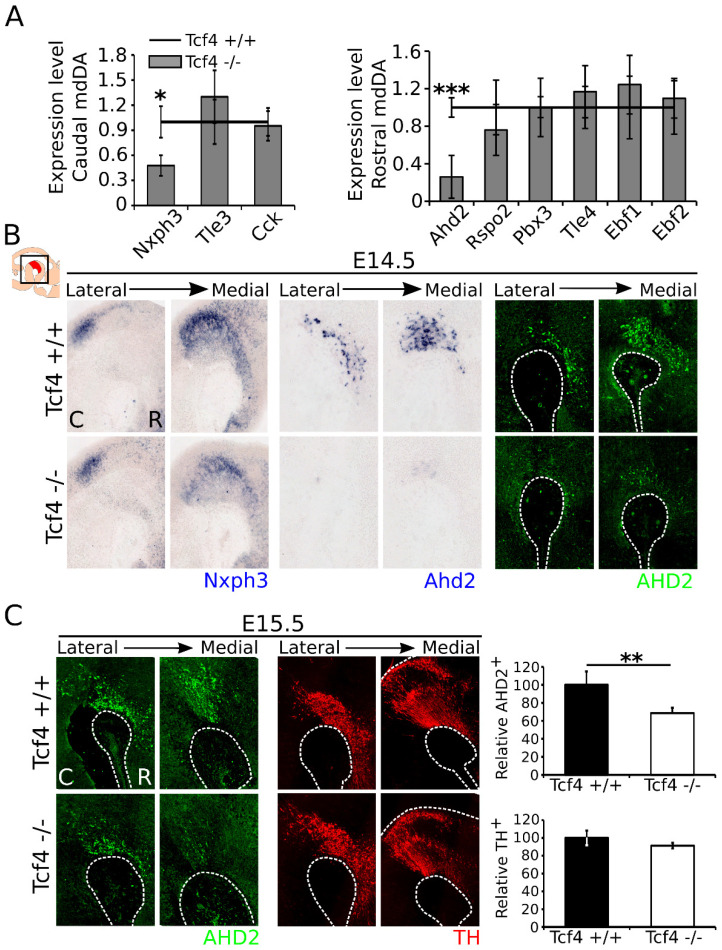
Expression of the caudal marker *Nxph3* and the rostral marker *Ahd2* is decreased in E14.5 *Tcf4* mutant embryos, although expression of AHD2 partly recovers at E15.5. (**A**) qPCR of caudal and rostral markers in the mdDA system. Expression of caudal markers *Tle3* and *Cck* is unaffected in the *Tcf4* mutant (n = 4; two-tailed). Expression of *Nxph3,* on the other hand, shows a significant decrease of ~50% (* n = 4; *p* = 0.036 two-tailed). Expression of rostral markers *Rspo2*, *Pbx3*, *Tle4*, *Ebf1*, and *Ebf2* is not altered in the mutant (n = 4; two-tailed). However, expression of the rostral marker *Ahd2* is decreased with ~75% in the mutant (*** n = 4; *p* = 0.0008 two-tailed). WT expression levels were set at 1. (**B**) In situ hybridization of *Nxph3* (blue, left panel) and *Ahd2* (blue, right panel) confirms the qPCR data and shows that these markers are visibly altered in the midbrain of the *Tcf4* mutant compared to WT littermates. AHD2 protein (green) expression is similarly affected in the *Tcf4* mutant midbrain at E14.5. C: caudal, R: rostral. (**C**) At E15.5, the expression of AHD2 (green) is partly recovered. Quantification shows an ~30% decrease in the amount of AHD2^+^ cells at E15.5 (** WT n = 3, mutant n = 4; *p* = 0.006 two-tailed), although the expression of TH is not visibly altered and the amount of TH^+^ cells is not significantly changed at this stage (black bars) (WT n = 3, mutant n = 4; one-tailed). WT was set at 100%. C: caudal, R: rostral.

**Figure 4 biomedicines-09-00317-f004:**
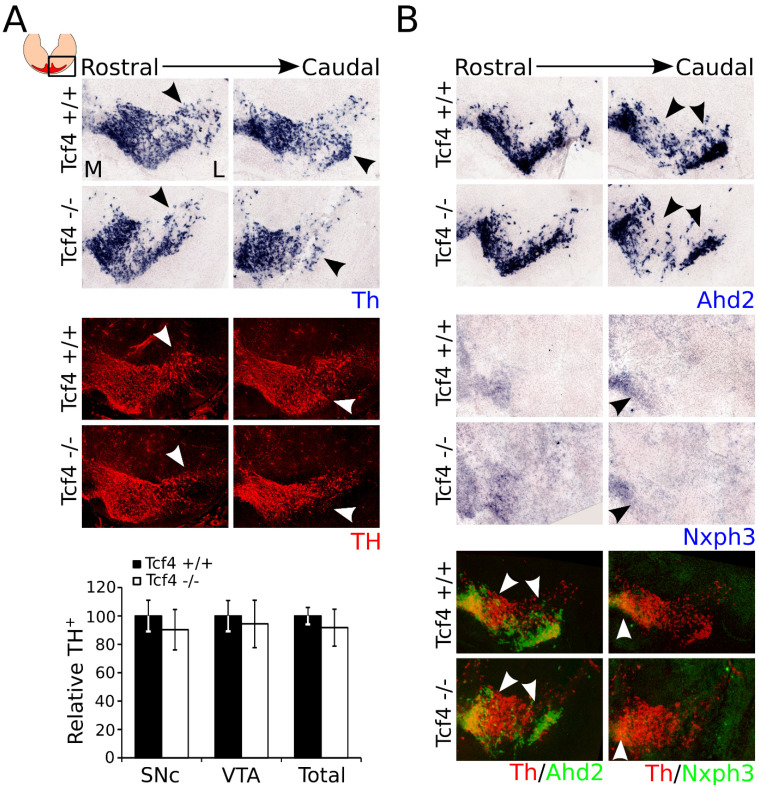
Expression of *Th* is affected in the mutant midbrain at P0, whereas the expression of *Ahd2* and *Nxph3* is partially recovered. (**A**) Expression of *Th* transcript (blue) in the mutant midbrain at P0 shows a decrease in the most lateral parts of the mdDA neuronal population (black arrowheads). Immunohistochemistry for TH-protein (red) shows a similar phenotype (white arrowheads). However, quantification of the TH-expressing neurons in the SNc, VTA, and the total population does not show a significant downregulation in the mutant (white bars) compared to the WT (black bars) (n = 3, two-tailed) M: medial, L: lateral. (**B**) *Ahd2* (blue) expression is mostly recovered, but still shows a decrease in expression in the caudal regions of the mdDA neuronal population (black arrowheads). *Nxph3* (blue) expression is similarly mostly recovered, but still shows a decrease in expression in the caudal regions of the mdDA neuronal population (black arrowheads). Pseudo-overlays of adjacent sections of *Th* (red) with *Ahd2* (green) and *Nxph3* (green) shows that the expression of *Ahd2* and *Nxph3* coincides with the expression of *Th*, and the affected areas are similar between the different markers (white arrowheads).

**Figure 5 biomedicines-09-00317-f005:**
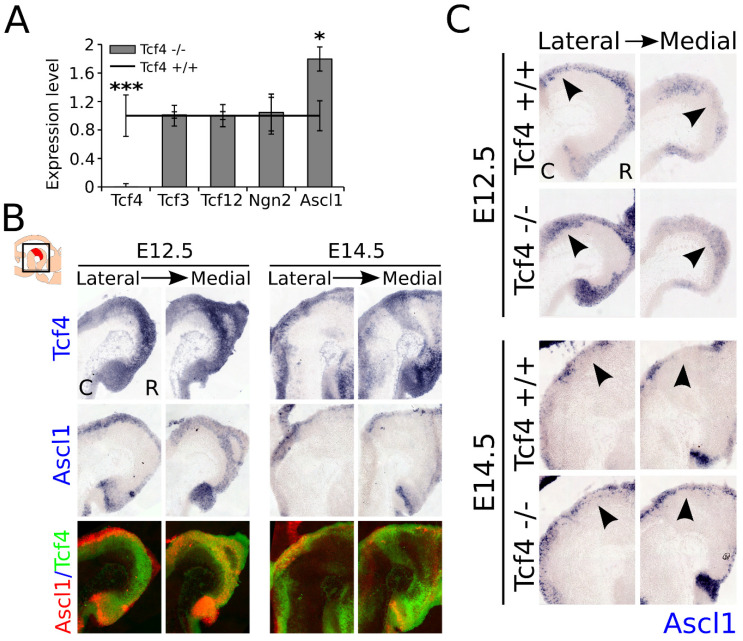
*Ascl1* expression is upregulated in the *Tcf4* mutant at E12.5 and E14.5. (**A**) The qPCR of E14.5 *Tcf4* mutant and WT dissected midbrain material shows that in the mutant, the deleted part of the genome is indeed lost in the *Tcf4* transcript (*** n = 4; *p* = 0.001 two-tailed). Expression levels of different bHLH factors, *Tcf3*, *Tcf12*, *Ngn2*, and *Ascl1*, show that loss of *Tcf4* leads to an upregulation of *Ascl1* in the mutant midbrain at E14.5 (* n = 4; *p* = 0.05 two-tailed), whereas expression of the other bHLH factors in the midbrain remains unaffected. WT levels were set at 1. C: caudal, R: rostral. (**B**) WT expression of *Tcf4* (blue) and *Ascl1* (blue) shows an overlap in expression of areas at E12.5 and E14.5 in the VZ of the midbrain. However, at E14.5 the expression of *Ascl1* appears to be absent in areas where *Tcf4* is strongly expressed, which is clearly seen in the pseudo-overlays of the expression areas of *Tcf4* (green) and *Ascl1* (red) in adjacent sections. C: caudal, R: rostral. (**C**) In situ hybridization of *Ascl1* (blue) on E12.5 and E14.5 embryonic midbrain shows an expanded area of expression of *Ascl1* in the VZ of the embryonic midbrain in the mutant (black arrowheads). C: caudal, R: rostral.

## Data Availability

The data presented in this study are available on request form the corresponding author(s).

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
