# Peer review of "Tcf4 Is Involved in Subset Specification of Mesodiencephalic Dopaminergic Neurons"

_biomedicines, 2021, doi:10.3390/biomedicines9030317_

Round 1

Reviewer 1 Report

Please see the attached review report.

Author Response

MDPI, Biomedicines

Tcf4 affects subset specification in mesodiencephalic dopaminergic neurons

Simone Mesman, Iris Wever and Marten P. Smidt

December 2020

Review Report

Mesman S. et. al., assess the role of Tcf4, a transcription factor belonging to the E-box sub-family of bHLH proteins, in the development of mesodiencephalic dopaminergic (mdDA) neuronal system in the embryonic mouse midbrain. To address this question, they compare the expression of different mdDA neuronal markers at the mRNA and protein level between Tcf4 +/+ (wildtype) with Tcf4 -/- (mutant) embryos from embryonic day E11.5 – E15.5. In Tcf4 WT mice they show that Tcf4 RNA is expressed at E11.5 - E15.5 and not anymore at postnatal day (P) 0 in the ventricular zone (VZ) and overlaps with the rostral part of the neuronal population expressing the dopaminergic neuron-specific marker tyrosine hydroxylase (TH) in this region. In Tcf4 mutants they observe a significant increase in the number of TH expressing neurons at E11.5, which is normalized at E14.5. The authors suggest that this might be an accelerated differentiation of mdDA neurons and could affect the sub-type specification of these cells. Moreover, in some Tcf4 mutants they observed at E14.5 a reduction of PITX3 in the midbrain, which was not seen at E13.5 and E15.5. The authors show a decrease in the mRNA expression of the sub-type specific markers Nxph3 (caudal marker) and Ahd2 (rostral) marker of this brain region. For AHD2 they observe a similar decrease in protein level at E14.5 and E15.5. At P0 they find that TH mRNA and protein expression levels are not significantly different from WT and Nxph3 and ahd2 transcript levels are comparable to WT, with slight region-specific differences. Mesman S. et. al., further show that in Tcf4 mutant embryos there is an increase in mRNA expression of Ascl1, an E-box family member of Tcf4. The authors argue that the loss of Tcf4 might be compensated for by increased expression of Ascl1 and this results in the reported mild phenotype in mdDA neuronal differentiation in Tcf4 mutants.

The work presented in this manuscript has to be extended with more data in order to support the conclusions presented in this manuscript. Please find my specific comments below.

Major Comments

  1. The authors assess whether Tsf4 plays a role in sub-type specification of the TH+ neuronal population in the midbrain. They find that the TH+ transcript and protein is lower in mutants compared to WT at E11.5, but then recovers. At P0 there seems to be a sub-region specific reduction in TH mRNA and protein. Transcripts of the subset markers Nxph3 (caudal region) and Ahd2 (rostral region) are downregulated at E14.5 in Tcf4 mutants, but then also recover with slight sub-region specific differences at P0. What is in general missing in this manuscript is a double labelling of the TH+ population and the caudal and rostral markers. This is needed to clearly show that these markers are in these mutants expressed in the TH+ population and whether the pattern in this neuronal population changes. Especially in Figure 4, where the arrowheads point to specific regions where it seems that the expression of these markers is different between WT and mutants. Does the region where TH seems less expressed overlap with the region where Nxph3 and Ahd2 are also less expressed? This you need to show in order to say that Tcf4 regulates the sub-type specification of the TH+ population by regulating the expression of Nxph3 and Ahd2.

Answer: We agree with the reviewer that overlapping expression patterns of the different markers would give more insight in the effects of the loss of Tcf4. Therefore, we have added pseudo-overlays of the expression patterns of Th and Ahd2, and Th and Nxph3 in adjacent sections to the figures to provide more clarity. Furthermore, it is known from previous studies from our and other groups that Ahd2 is a marker for the rostral population and Nxph3 is a marker for the caudal population. These references have been added to the text.

The authors also find an expansion of the Ascl1 transcript labelling at E12.5 and E14.5 in the midbrain. They suggest that, since this is another E-box family protein, its increased expression in Tcf4 mutants might compensate for the loss of Tcf4. Also here, to suggest this the authors have to show that the specific brain region where Ascl1 is aberrantly expressed in the mutant’s overlaps with the regions where normally Tsf4, Nxph3 and Ahd2 are expressed.

Answer: We agree with the reviewer and have added the WT expression patterns of Ascl1 and Tcf4 at E12.5 and E14.5. Pseudo-overlays of the expression patterns of adjacent sections clearly show overlapping expression patterns at early stages and more exclusive expression patterns at later developmental stages.

  1. Line 188: “…., suggesting an accelerated neuronal differentiation at E11.5”

To suggest an accelerated neuronal differentiation, you have to show the development of this population across time and not only compare E11.5 and the final time point E14.5, where already no difference is anymore between mutant and WT. To understand whether it is an accelerated neuronal differentiation you should determine when the full TH+ population is present in mutants compared to WT. E14.5 is too late, because there is already no difference. Potentially in mutants the population present at E14.5 is already present at E13.5 and in WT only at E14.5. This better shows you that there is an accelerated neuronal differentiation. Or in the mutants the first TH+ neurons might already appear before E11.5. Without this information it can be for the same a mis-specification of the neuronal population or a higher total neuronal number which causes an increase in the TH+ population at E11.5.

Answer: We agree with the reviewer that it is difficult to determine an accelerated differentiation of TH-expressing neurons. We have added E12.5 TH-expression data of WT and mutant animals in a supplemental figure to provide more clarity on this matter and adjusted the text to make a milder statement regarding the increase in TH-expressing neurons at E11.5.

  1. Line 195: “……Loss of PITX3 in the midbrain,…”

It is not explained what factor PITX3 is. This has to be done.

Answer: We have added an explanation about the factor PITX3 to the text.

PITX3 loss was only observed in some mutant embryos at E14.5, while in others there was no difference. What means some? 1 out of 3? 2 out of 3? When it is just one animal, then this data should be either removed from this paper, because it only adds confusion and no constructive data or the “n” should be increased.

Answer: We have decided to keep the image in the manuscript, as we feel that this adds to the overall insight in the function of Tcf4 in midbrain development. However, we did change the text and clarified the amount of embryos in which this effect was detected.

Line 197 and 210, 211: “The effect on PITX3 was only observed at E14.5 and not E13.5 and

E15.5”.

When in Figure 2C the images displaying staining of PITX3 in the lateral section at E15.5 are compared between Tcf4 +/+ and Tcf4 -/- then PITX3 staining seems to be clearly reduced in the -/- section. Why is this stated as not different?

Answer: After careful reconsideration of the data, we still deem the expression of PITX3 in E15.5 to not differ between the WT and mutant embryos. Therefore, the text has not been changed.

When PITX3 has an effect on the terminal differentiation of the TH+ population, does the expression of these factors overlap? In the mutants where PITX3 expression is normal (not different from WT), is in these mutants also the TH+ population/expression normal?

Answer: As stated in the text, the TH+ expression was not affected in the mutants at E14.5, also not in the mutants that showed an effect on PITX3 expression. Normally PITX3 is expressed in post-mitotic dopamine neurons, similar to TH. For us this it was striking that there was no change in TH expression in these animals, but this is not implausible and is known to happen in other mutants as well. For instance in the Tcf12 mutant (Mesman and Smidt 2017).

  1. Line 221 – 222: “….., whereas expression of Nxph3 shows a decrease……..(Veenvliet,

unpublished)”.

Does this mean that this data is not produced in the context of the project described in this manuscript? Is this data obtained from samples not collected from the mice analysed in the current study? This should be more clearly defined.

Answer: We agree with the reviewer that this should be more clearly defined. The data was generated for this paper. However, the fact that Nxph3 is a caudal marker was not previously published and detected by a former colleague and personal communication. Recently, a paper was published showing this marker for caudal neurons, so we have removed the Veenvliet reference and refered to the correct paper.

  1. Why is only the protein level from AHD2 shown and analyzed and not the one from Nxph3?

There are only these two factors which are changed in Tcf4 mutants compared to WT and thus to better support the conclusion that Tcf4 is regulating sub-type specification of mdDA neurons it is important that the decrease in the caudal mdDA marker Nxph3 can be shown both at transcript and protein level.

Answer: Unfortunately, we do not have a working antibody for Nxph3. Therefore, we decided to perform immunostainings for AHD2 and only in situ hybridization for Nxph3.

  1. Line 273: “….., suggesting a role of Tcf4 in survival of these neurons”.

In order to suggest a role in survival a staining for the total neuronal population should be done and the percentage of TH+ cells determined. Without this data it is too strong to suggest a loss of neurons.

It might be a loss of neurons, but maybe also a loss of the marker. TH+ neurons are at E11.5 higher in mutant, at E14.5 they show similar numbers as WT and P0 there seems to be a small reduction. Other markers which are important for rostral caudal mdDA sub-type specification are altered at E14.5 in Tcf4 mutants. Is at this time-point the TH population already fully present are still being formed? In the latter case, in the mutant there might be still a second effect on TH neuron specification.

Answer: We agree with the reviewer that we cannot state a role in survival of TH-neurons for Tcf4 and that it might as well be that the marker is simply lost, but the neurons are still present. Therefore, we have adjusted the text to make a milder claim on the effects of Tcf4 on the TH-expressing neuronal population.

Minor comments:

  1. Line 88 – 90: ….”mutant embryos were generated by crossing heterozygous Tcf4 mice. WT embryos, to study the WT expression pattern of Tcf4, were generated by crossing C57BL/6 mice”.

For the expression of some markers, quite a bit embryo to embryo variability was observed. This might have been less when littermates would have been used as controls. Why was this not done?

Answer: We used WT and mutant littermates to determine differences in expression of markers in the mdDA neuronal population. C57BL/6 animals were used to gather the data as seen in Figure 1 and for the WT expression of Tcf4 and Ascl1 in figure 5B. We have adjusted the text to clarify this.

  1. Line 161 – 162: “At E14.5 the expression of Tcf4…….to the rostral area in both lateral and medial sections (Figure 1a and 1b)”.

Should here not be only referred to Figure 1A? In Figure 1B I don’t see a difference between rostral and caudal in terms of Tcf4 expression region. When, then it is even more intense in the caudal section.

Answer: We have adjusted the text to clarify where Tcf4 is expressed and correctly referred to Figure 1B .

  1. Figure 1, legend, line 171: “Table 4”

This is a typo and probably should be Tcf4.

Answer: We have adjusted the text accordingly.

  1. Line 172, 173: “…., which is more apparent at E14.5 and even strongly overlaps with the rostral TH+ population”. Rostral and caudal is not indicated in the images in Figure 1A, this should be done.

Answer: We have adjusted the figures so that Rostral and Caudal, Medial and Lateral parts are correctly indicated in all figures.

  1. Line 186, 187: “In the Tcf4 mutant midbrain……….compared to mutant embryos”…. “mutant embryos” should probably be WT embryos.

Answer: We have adjusted the text accordingly

  1. Figure 3: A) In the graphs you cannot discriminate WT from mutant error bars. These graphs have to be changed in order to clearly visualize the error bars of these two groups separately.

Is the data plotted in the graph showing the expression level of caudal mdDA markers obtained from mice described in this study or from another study (Veenvliet)? This is not clear from the text.

Answer: We have adjusted the text accordingly.

  1. Figure legend 3C: The text is very confusing: “AHD2 protein (green) expression is similarly affected as Ahd2 transcript expression in Tcf4 mutant midbrain (white bars) at E14.5 compared to the WT (black bars). At E15.5 on the other hand the……..Quantification shows an ~30% decrease in the……at E15.5” There is only 1 graph displayed in Figure 3C showing AHD2 levels. Does the data from this graph come from E14.5 or E15.5 sections?

Answer: We have adjusted the figure and the accompanying legend and description in the text accordingly so that it is clear as to where the data is obtained from.

  1. Line 266 – 270: “Nxph3 shows a weak expression……still slightly affected…..” This sentence is very confusing. Does the first part of the sentence refer to expression in the WT? This should be indicated.

Answer: We have adjusted the text accordingly.

  1. Line 303 - Typo: “….Tcf4 results in an increase is expression….” Has to be: an increase in expression.

Answer: We have adjusted the text accordingly

  1. There is an inconstancy between the labelling of the Figures (capitals: A, B) and the reference to the figures in the main text (a, b).

Answer: We have adjusted the text accordingly

Reviewer 2 Report

The authors demonstrated the novel physiological function of Tcf4 for the developmental regulation of dopaminergic phenotype. The manuscript is clear, concise, and well-written.

Minor point:

Several E-box motifs are present in human, mouse, and rat TH promoters. Do you have any idea that Ncf4 binds to the TH promoter region to activate or repress the promoter activity developmentally in mice? Please discuss to include or exclude the possibility.

Author Response

Reviewer 2

The authors demonstrated the novel physiological function of Tcf4 for the developmental regulation of dopaminergic phenotype. The manuscript is clear, concise, and well-written.

Minor point:

Several E-box motifs are present in human, mouse, and rat TH promoters. Do you have any idea that Ncf4 binds to the TH promoter region to activate or repress the promoter activity developmentally in mice? Please discuss to include or exclude the possibility.

Answer: We have added this point to the discussion so that this is discussed. We do not know whether Tcf4 binds to the promoter of the Th gene in the murine midbrain, but considering the minor effect on Th-expression we do not deem this binding to be present or to be very important in regulation of Th expression in the mouse brain.

Round 2

Reviewer 1 Report

The modifications to the text and figures have clearly improved the overall quality of the manuscript. The conclusions as formulated in this adjusted manuscript are justified. I support publishing this manuscript in the current form.